# FAM83A and FAM83B as Prognostic Biomarkers and Potential New Therapeutic Targets in NSCLC

**DOI:** 10.3390/cancers11050652

**Published:** 2019-05-11

**Authors:** Sarah Richtmann, Dennis Wilkens, Arne Warth, Felix Lasitschka, Hauke Winter, Petros Christopoulos, Felix J. F. Herth, Thomas Muley, Michael Meister, Marc A. Schneider

**Affiliations:** 1Translational Research Unit, Thoraxklinik at Heidelberg University Hospital, D-69126 Heidelberg, Germany; sarah.richtmann@med.uni-heidelberg.de (S.R.); thomas.muley@med.uni-heidelberg.de (T.M.); michael.meister@med.uni-heidelberg.de (M.M.); 2Translational Lung Research Center Heidelberg (TLRC), Member of the German Center for Lung Research (DZL), D-69120 Heidelberg, Germany; hauke.winter@med.uni-heidelberg.de (H.W.); petros.christopoulos@med.uni-heidelberg.de (P.C.); felix.herth@med.uni-heidelberg.de (F.J.F.H.); 3Microbial Energy Conversion and Biotechnology, Department of Biology, Technische Universität Darmstadt, D-64287 Darmstadt, Germany; wilkens@bio.tu-darmstadt.de; 4Institute of Pathology, Heidelberg University Hospital, D-69120 Heidelberg, Germany; warth@patho-uegp.de (A.W.); email@felixlasitschka.de (F.L.); 5Department of Surgery, Thoraxklinik at Heidelberg University Hospital, D-69126 Heidelberg, Germany; 6Department of Thoracic Oncology, Thoraxklinik at Heidelberg University Hospital, D-69126 Heidelberg, Germany; 7Department of Pneumology and Critical Care Medicine, Thoraxklinik at Heidelberg University Hospital, D-69126 Heidelberg, Germany

**Keywords:** NSCLC, FAM83A, FAM83B, biomarker, EGFR-TKI

## Abstract

Although targeted therapy has improved the survival rates in the last decade, non-small-cell lung cancer (NSCLC) is still the most common cause of cancer-related death. The challenge of identifying new targets for further effective therapies still remains. The FAMily with sequence similarity 83 (FAM83) members have recently been described as novel oncogenes in numerous human cancer specimens and shown to be involved in epidermal growth factor receptor (EGFR) signaling. Here, gene expression of *FAM83A* and *B* was analyzed in a cohort of 362 NSCLC patients using qPCR. We further investigated relations in expression and their prognostic value. Functional assays in NSCLC cell lines were performed to evaluate *FAM83A* and *B* involvement in proliferation, anchorage-independent growth, migration, and the EGFR pathway. We observed a highly increased gene expression level of *FAM83A* (ø = 68-fold) and *FAM83B* (ø = 20-fold) which resulted in poor survival prognosis (*p* < 0.0001 and *p* = 0.002). Their expression was influenced by EGFR levels, pathway signaling, and mutation status. Both genes affected cell proliferation, and *FAM83A* depletion resulted in reduced migration and anchorage-independent growth. The results support the hypothesis that *FAM83A* and *B* have different functions in different histological subtypes of NSCLC and might be new therapeutic targets.

## 1. Introduction

Lung cancer is the most commonly diagnosed cancer and the leading cause of cancer-related death worldwide [1]. While a steady increase in survival is observed for most cancer types, advances have been slow for lung cancer which is typically diagnosed at an advanced stage, with a five-year survival rate of only 5% [2]. Non-small-cell lung cancer (NSCLC) accounts for 85% of all lung cancers and is further classified into the main histological groups large-cell neuroendocrine carcinoma (LCNEC), squamous cell carcinoma (SQCC), and adenocarcinoma (ADC) [3]. Personalized medicine and the identification of molecular targets in tumors represent a promising field of novel therapy approaches. Patients treated with inhibitors of epidermal growth factor receptor (EGFR) or anaplastic lymphoma kinase (ALK) mutant proteins have shown improved survival [4]. In NSCLC, activating mutations in the EGF receptor account for about 15% of the mutations in adenocarcinoma. Tyrosine kinase inhibitors (TKIs) such as gefitinib and erlotinib or monoclonal antibodies are approved drugs for the treatment of locally advanced or metastatic NSCLC harboring EGFR mutations [5]. Nevertheless, the plasticity of cancer cells and the compensatory activation of downstream pathways often result in resistance to targeted therapies [6,7]. Consequently, new oncogenes need to be found showing significantly increased expression in cancer cells to circumvent resistance and to access alternative proliferation and cell growth cascades.

The members of the FAMily with sequence similarity 83 (FAM83) have been discovered to be upregulated in a variety of cancer specimens with the potential to serve as new targets [8,9]. The protein family is characterized by a highly conserved domain of unknown function (DUF1669) at the N-terminus, while the C-terminal regions vastly differ in the different family members [10]. A recent study could determine isoforms of the casein kinase 1 (CK1) family as interactors of the DUF1669 of FAM83 proteins [11]. FAM83A and B proteins have been identified as possible key regulators in the EGFR pathway in breast cancer cells, leading to the development of resistance to TKIs [7,12,13]. However, the biological role of FAM83A and B in cancer cells, especially in NSCLC, still remains unclear, while a detailed understanding is critical to develop novel therapeutic approaches.

In the current study, we investigated the expression and function of *FAM83A* and *B* in NSCLC in general and in relation to the EGFR pathway. The analyses revealed that both genes might serve as effective new diagnostic and prognostic biomarkers in NSCLC, with severe impact on various biological functions.

## 2. Results

### 2.1. FAM83A and B Are Highly Overexpressed in NSCLC

To investigate their potential diagnostic properties in NSCLC, gene expression of *FAM83A* and *FAM83B* was analyzed by qPCR in a cohort including 362 patients (Table 1).

*FAM83A* was overexpressed in the tumor tissue in 90% of all patients, reaching up to a 10,000-fold amount of mRNA compared to non-neoplastic tissue (Figure 1A). Besides, we observed differences between ADC (92-fold overexpression) and SQCC (26-fold overexpression) (Figure 1B). For *FAM83B*, a median 20.23-fold overexpression was determined (Figure 1D), with an even larger difference between expression in ADC and expression in SQCC (Figure 1E). In ADC tumor tissue, the *FAM83B* gene was overexpressed with a 3.76-fold median. In contrast to this, the data regarding SQCC revealed a 109.8-fold expression. A further subdivision of the data into male and female patients did not reveal major differences in *FAM83B* expression (Figure 1F), while a higher median *FAM83A* level was detected in male patients with ADC compared to females (Figure 1C). In line with the observations of the expression patterns, a correlation analysis revealed that *FAM83A* expression did not relate to *FAM83B* expression in thedistinct histologies (Table 2). These data emphasize the hypothesis that the two genes have different roles in NSCLC.

### 2.2. FAM83A and FAM83B Are Prognostic Markers for NSCLC

To get an insight into the prognostic value of *FAM83A* and *B* gene expression, a univariate analysis was used to estimate the effect of high versus low expression on the overall survival of the patient cohort. The cut-off value to separate the whole cohort into two groups was calculated with a biomathematical tool for each gene (see material and method section). Both genes had a significantly impaired effect on the prognosis of the patients, represented by increased hazard ratios (*FAM83A*: HR 1.965 (1.366–2.817), *FAM83B*: 1.808 (1.252–2.612) Table 3). A multivariate analysis confirmed that *FAM83A* and *B* seem to be robust prognostic markers, with a highly significant negative effect on the overall survival of the patients (Table 3). Kaplan–Meier plots were used to visualize the overall survival over time. Corresponding to the data from Cox regression, a clear prognostic effect was seen for the two genes (Figure 2A,B). Furthermore, the expression profile of the genes was investigated depending on histology (Figure 2C–F) and gender (Figure 2G–J). High *FAM83A* expression in general led to poor survival in all patients, while the outcome was worse in patients with ADC (*p* = 0.002, Figure 2C) compared to patients with SQCC (*p* = 0.013, Figure 2E). While the median overexpression of *FAM83B* in ADC was relatively low (Figure 1E), elevated *FAM83B* tumor expression had a highly significant effect on the outcome (Figure 2D). A low expression of *FAM83B* in SQCC was only identified in 10 patients, thus the data must be interpreted with caution (Figure 2F). Increased *FAM83A* expression affected prognosis negatively for both genders, with a higher effect in females (*p* < 0.0001, Figure 2G,I). The survival rate after five years was approx. 90% for females with a low gene expression compared with about 50% for females with a high *FAM83A* expression. In contrast, the expression level of *FAM83B* showed a significant negative impact in males but not in females (Figure 2H,J).

### 2.3. The Expression Levels of FAM83A and B Are Related to EGFR Expression In Vitro and In Vivo

Previous studies in breast cancer cells have identified FAM83A and B as interactors of different components within the EGFR pathway, like PI3K or RAF [9,13]. Within this context, several experiments have indicated that they mediate resistance to different inhibitors targeting the EGFR or downstream effectors such as Akt and mTOR [14]. In NSCLC, driver mutations in the *EGFR* gene have been found in about 15% of all ADCs [5,15]. Therefore, the expression of *FAM83A* and *B* was analyzed in tumor tissue of NSCLC patients with ADC expressing either wild-type EGFR (79 patients) or EGFR signaling activating mutations (29 patients, Table 4). In contrast to other studies, the overexpression of either *FAM83A* or *FAM83B* in tumors harboring mutant EGFR could not be confirmed (Figure 3A,C). Conversely, *FAM83A* expression was significantly higher (*p* < 0.005) in patients with wildtype EGFR, while for *FAM83B,* no significant difference was observed. A correlation analysis was performed focusing on the relation between *EGFR* expression in the histological types ADC and SQCC and *FAM83A* and *B* expression within the patient cohort depicted in Table 1 (Figure 3B,D). Interestingly, the analysis revealed that in SQCC, the levels of *FAM83B* and *EGFR* slightly correlated (*r* = 0.51) (Figure 3D). For *FAM83A*, a correlation with *EGFR* failed to be relevant (Figure 3B). The results were underlined by an approach in 12 NSCLC cell lines in which the absolute copy number of both genes was estimated (Figure 3E). Here, the copy numbers of *FAM83B* and *EGFR* in SQCC cell lines highly correlated (*r* = 1). Besides, the highest amount of *FAM83A* mRNA in ADC cell lines was found in H1975. This cell line harbors a L858R/T790M double mutation in the *EGFR* gene, suggesting a potential impact on *FAM83A* expression in this cell line. These findings indicate a relation between the two *FAM83* genes and the expression level of *EGFR*.

### 2.4. Influence of FAM83A and FAM83B on Biological Functions in NSCLC Cell Lines

Proliferation, the ability of migration, and anchorage-independent growth (AIG) are essential characteristics of tumor cells. The involved pathways are highly regulated. The identification of specific oncogenes which participate in these functions represents an emerging field in targeted therapy research. As both *FAM83A* and *B* were significantly overexpressed in NSCLC, their biological function was investigated in several NSCLC cell lines to obtain insight into their role in cancer cells. Therefore, the genes were downregulated using siRNA-mediated depletion, and the effects on cell viability, migration, and AIG were evaluated (Figure 4). As *FAM83A* was mainly overexpressed in patients with lung adenocarcinoma, ADC cell lines expressing different amounts of *FAM83A* were analyzed. A549 is a wild-type-EGFR cell line that showed lower expression of *FAM83A* compared to the other used cell lines H1975 and H2228 (Figure 3E). H1975 cells harbor EGFR mutations, while H2228 contains a wild-type EGFR but also an *EML4-ALK* rearrangement. Since *FAM83B* was observed to be highly overexpressed in patients with SQCC, analyses were performed in the SQCC cell lines H1869, 2106T, and 161735T. The primary cell line 161735T was confirmed to be derived from squamous epithelium by the expression of the specific markers p63, cytokeratin 5/6, and cytokeratin 14 [16,17] (Appendix A). Downregulation of *FAM83* expression was confirmed by qPCR (Figure 4A for *FAM83A* and Figure 4E for *FAM83B*), and single siRNAs were pooled. The knockdown efficacy of the single siRNAs is shown in Appendix A. It was not possible to confirm the knockdown by western blot because of the lack of specific commercial antibodies. Moreover, producing specific antibodies for the current project failed, since the obtained antibodies appeared not to be suitable to detect transiently overexpressed FAM83A and B in different cell lines [18]. *FAM83A* depletion led to a highly significant decrease of all investigated properties. Only in H2228, proliferation was less restricted than in other cell lines, and an effect on AIG was not observed (Figure 4B,D). In the SQCC cell lines, an impact of *FAM83B* could not be confirmed, except for a slight negative influence on proliferation. The highest knockdown efficiency among the SQCC cell lines was achieved in the tumor-derived cell line 161735T (Figure 4E). Here, the ability of AIG was impaired upon *FAM83B* depletion, while proliferation was poorly affected (Figure 4F,H). As the cells did not migrate through the pores of transwells, the influence of *FAM83B* expression could not be determined (Figure 4G). The results indicate that both *FAM83A* and *B* are involved in cell proliferation and that *FAM83A* also regulates migration and AIG.

### 2.5. Impact of TKI Treatment and EGFR Inhibition on FAM83A and B Gene Expression and Cell Viability

In addition to the assays analyzing the biological function of *FAM83A* and *B* in NSCLC, the relation of *FAM83A* and *B* to EGFR signaling was investigated in the ADC cell lines H1975 and HCC827. H1975 cells harbor a L858R/T790M double mutation in the *EGFR* gene which is known to mediate resistance to first-generation TKIs. HCC827 cells showed the highest *EGFR* copy number and contain an exon 19 deletion leading to enhanced activity of EGFR. The cells were treated with the TKIs erlotinib and AZD9291 (osimertinib) that block the phosphorylation of EGFR kinase domain. Consequently, downstream signaling pathways are adversely affected, resulting in reduced proliferation and tumorigenicity. Erlotinib is a reversible small-molecule ATP analogue which has been found to be most effective in advanced NSCLC patients with a tumor containing exon 19 deletions or a L858R mutation in exon 21 of the gene encoding the EGFR kinase domain [19,20]. AZD9291 is an irreversible third-generation TKI, specific for mutant EGFR and especially targets the T790M resistance mutation [21]. As we observed a relation between the expression of EGFR and those of *FAM83A* and *B*, we further investigated whether they might be affected by EGFR-targeted TKI treatment in NSCLC cell lines (Figure 5). Previous studies have shown that FAM83A and B mediate resistance to TKIs in breast cancer cells [9,13,14]. Thus, we analyzed whether TKIs like erlotinib and AZD9291 influence *FAM83A* and *B* expression in NSCLC cell lines. First, cells were treated with 0.1 µM, 1 µM, and 10 µM AZD9291 and erlotinib, and gene expression of *FAM83A* and *B* was analyzed by qPCR. In H1975, evaluation of the data revealed a predominantly negative impact of either TKI treatment on both genes (Figure 5A). In HCC827, there was only a slight effect on *FAM83B* expression (Figure 5B). In contrast to this, *FAM83A* was significantly diminished. Regarding the general toxicity of the TKIs, cell numbers were evaluated upon treatment (Figure 5C,D). In H1975, treatment with 10 µM of either TKI led to the highest decrease of cell viability. AZD9291 had an overall more effective impact which is based on its characteristic to specifically target the T790M mutation found in H1975 (Figure 5C). In HCC827, both TKIs led to a cell number reduction of 40–50% regardless of the concentration (Figure 5D). To investigate the influence of *FAM83A* and *B* on drug sensitivity, the cells were treated with siRNA pools. The knockdown of the genes was very efficient and affected cell proliferation in both cell lines (Figure 5E,F). In contrast to the effect of *FAM83B* downregulation in SQCC cell lines, depletion of the gene in the ADC cell lines led to a significantly impaired cell viability (Figure 5E,F), although the absolute copy number was relatively low in these cells (Figure 3E). The siRNA-mediated depletion of *FAM83A* and *B* followed by TKI treatment of H1975 enhanced the negative impact of the TKIs on cell viability compared to the control (Figure 5G). The effect of erlotinib was not influenced by *FAM83B* depletion, though. Besides, knockdown of the two genes only led to a slightly greater reduction of the cell number in HCC827 (Figure 5H). These results indicate an additive effect of *FAM83A* and *B* downregulation and TKI treatment in NSCLC, leading to a higher cell toxicity. In compliance with the data from the previous experiments, *FAM83A* and *B* expression seems to be affected differently depending on the EGFR mutation status and EGFR regulation.

## 3. Discussion

Advanced-stage NSCLC remains largely incurable as a consequence of the high flexibility of cancer cells and multiple regulatory feedback loops resulting in therapy resistance [6]. Consequently, novel oncogenes that are highly overexpressed in tumor tissue and can serve as new targets have to be found. The members A and B of FAM83 were recently described to be highly upregulated in several cancer specimens [14]. Their function in cancer cells is largely unknown, and especially their role in lung cancer remains unclear. In order to acquire a better understanding of *FAM83A* and *B* in NSCLC and to assess their possible diagnostic and prognostic values, we investigated the genes regarding their expression patterns and involvement in biological functions. While *FAM83A* was upregulated in ADC, *FAM83B* expression was elevated especially in SQCC. These observations correspond to those of previous studies that could identify elevated amounts of the genes in tumor tissue of the respective histology [22,23]. Gene overexpression in general may indicate a tumor-promoting role, while variances in expression levels related to tumor subtypes suggest different roles and importance in the maintenance and tumorigenicity of the cancer cells. This might be underlined by the correlation analysis that showed poor relation between *FAM83A* and *B* in the two sub-histologies. Our multivariate analyses could further validate both genes as robust prognostic markers regarding the overall survival of the patients. Here, elevated expression of the genes affected survival negatively, and additional distinction of the patient cohort into the main histologies revealed a highly significant influence of *FAM83A* expression in ADC and SQCC, while *FAM83B* showed an impairment only in patients with ADC. In a recent study, high *FAM83B* expression was observed to relate to a better disease-free survival in SQCC, while an effect on overall survival was not observed [23]. In agreement with this, our analyses did not show an influence of *FAM83B* expression on the overall survival of patients with SQCC. However, instead of a rather positive effect, opposite results were obtained. As we chose the cut-off over all patients, only 10 patients were included for high *FAM83B* expression within SQCC, and the Kaplan–Meier plot did not result in an evaluable survival analysis. Nevertheless, when using the median gene expression in SQCC as a cut-off (as published in [23]), the results did not match those from the above-mentioned study [18]. The discrepancy might be based on the different methods used for the detection of *FAM83B* expression levels. While we measured gene expression by qPCR and normalized the values with respect to two housekeeper genes, in the study of Okabe et al. [23], immunohistochemical quantification was performed. Overexpression in tumor tissue was examined by western blot analysis, although the reliability of the applied antibody is uncertain for this type of analysis, as the size of the resulting single bands differ more than 20% from the predicted size and is not supported by experimental and/or bioinformatics data (Atlas Antibodies Cat# HPA031464, RRID: AB_10599973). Thus, the results appear questionable, while it might be taken into account that differences might also be due to the analysis of different ethnical groups (Asian and Caucasian) [24].

In HER2-positive breast cancer, resistance to the monoclonal anti-HER2 antibody trastuzumab has been correlated to *FAM83A* overexpression [12,25]. HER2 is amplified in about 20% of all breast cancer patients [26] and is a member of the same receptor tyrosine kinase family as EGFR which is known to be mutated in 15% of patients with lung ADC [15]. Because of its involvement in PI3K/Akt and MAPK signaling, *FAM83B* has also been shown to confer a decreased sensitivity to PI3K, Akt, and mTOR inhibitors in immortalized human mammary epithelial cells [13]. In lung ADC, *FAM83B* expression has been shown to be significantly elevated in tumors with wild-type EGFR [27], while in our patient samples, a significant difference was not observed. Here again, differences in quantification methods might be responsible for the divergent results. Moreover, the study from Yamaura et al. only included Japanese patients [24,27]. Interestingly, our analysis of lung ADC patients revealed that *FAM83A* had lower expression in tumors with a mutant variant of EGFR. However, it cannot be excluded that additional mutations in the tumor, like KRAS or BRAF, might influence *FAM83A* and/or *B* expression. Nonetheless, a correlation between *EGFR* expression and *FAM83A* and *B* was identified in SQCC patients as well as in cell lines. These data indicate a relation between the genes.

In addition to the evaluation of *FAM83* expression in tumor tissue, functional assays were performed in several NSCLC cell lines. In agreement with previous studies, *FAM83A* and *B* depletion had a significant effect on the proliferation and AIG of ADC cells [8,9,27]. We were able to show that downregulation of *FAM83A* resulted in impaired migration. Even in A549, which showed lower *FAM83A* expression compared to the other cell lines, depletion of the gene negatively affected all of the investigated cellular functions. This might indicate that *FAM83A* is highly regulated and that downregulation of the gene significantly impairs important properties in ADC cells with overall lower expression values, too. The effects on the cell line H2228, which contains an *EML4-ALK* rearrangement, were restricted to a significantly decreased number of migrating cells. In SQCC cells, a precise role of *FAM83B* could not be determined. Despite the high knockdown efficiency in 161735T, relatively high copy numbers were still detected in *FAM83B*-depleted samples. This might result in sufficient amounts of *FAM83B* in the cells to maintain major regulation mechanisms. In this case, a more effective downregulation of *FAM83B* might lead to more conclusive results, as a slight tendency of adversely affected proliferation and AIG was observed. The two ADC cell lines H1975 and HCC827 were further analyzed regarding the relation between *FAM83A* and *B* expression and EGFR inhibition and regarding possible effects of TKI treatment on the gene expression of *FAM83A* and *B*. Our data lead to the assumption that the impact of TKI treatment on gene expression highly depends on the mutational variant of EGFR. Further knockdown approaches revealed that only in H1975 cells an additive effect on proliferation could be observed. As in HCC827 cells *FAM83A* expression was already significantly decreased by TKI treatment, the additional knockdown did not result in further changes regarding cell viability. Nonetheless, validation in a larger patient cohort is mandatory to clarify the relation between *FAM83A* and *B* expression and the EGFR mutation status. In general, patients who undergo surgery are not tested for molecular alterations when the tumor is completely resected. Only surgical patients developing metastasis at a later time or patients with recurrence of tumor disease are retrospectively analyzed for EGFR mutations. Consequently, sufficient tissue amounts of patients with EGFR mutations are rarely available. Sampling material suitable for RNA analysis from stage IV patients is difficult to obtain, as well. Thus, collection of cryoconserved biopsies in inoperable patients might circumvent these obstacles to investigate the prognostic value of *FAM83A* and *B* in late stages.

Our work demonstrate that both *FAM83A* and *B* have high potential to serve as diagnostic and prognostic biomarkers in NSCLC. Their function and impact seem to be strongly related to tumor histology and EGFR expression and signaling. Future studies shall clarify more in detail the involvement of FAM83A and B in EGFR signaling in NSCLC. Therefore, further approaches should also focus on possible effects of *FAM83A* and *B* expression on the status of EGFR activation. Interaction studies might give hints on binding partners playing a role in developing resistance to targeted EGFR therapies. Since overcoming of resistance will be a major key for the enhancement of the survival rates in NSCLC, FAM83A and B are highly potential targets for intervention and development of effective therapeutic approaches.

## 4. Materials and Methods

### 4.1. Tissue Sample Collection, Characterization and Preparation

Tissue samples were provided by the Lung Biobank Heidelberg, a member of the accredited Tissue Bank of the National Center for Tumor Diseases (NCT) Heidelberg, the BioMaterialBank Heidelberg, and the Biobank platform of the German Center for Lung Research (DZL). The use of biomaterial and data for this study was approved by the local ethics committee of the Medical Faculty Heidelberg (S-270/2001). All patients included in the study signed an informed consent. Tumor and matched distant (>5 cm) normal lung tissue samples from NSCLC patients who underwent resection for primary lung cancer at the Thoraxklinik at University Hospital Heidelberg, Germany, were collected between 2003 and 2015. All diagnoses were made according to the 2004 WHO classification for lung cancer by at least two experienced pathologists [28]. Tumor stage was designated according to the 7th edition of the UICC tumor, node, and metastasis [29]. Tissues were snap-frozen within 30 minutes after resection and stored at −80 °C until the time of analysis. For nucleic acid isolation, 10–15 tumor cryosections (10–15 µm each) were prepared for each patient. The first and the last sections in each series were stained with hematoxylin and eosin (H&E) and were reviewed by an experienced lung pathologist to determine the proportions of viable tumor cells, stromal cells, normal lung cell cells, infiltrating lymphocytes, and necrotic areas. Only samples with a viable tumor content ≥50% were used for subsequent analyses. In our manuscript, we investigated two different patient cohorts: the main cohort consisting of 362 patients (Table 1), and the EGFR cohort including 108 patients (Table 4). The EGFR cohort partly overlapped with the main patient cohort but also included several other patients.

### 4.2. Cultivation of Cells

Cells were cultivated under humidified conditions at 37 °C and 5% CO_2_. The adenocarcinoma cell lines A549 (ATCC^®^ CCL-185™) and H1975 (ATCC^®^ CRL-5908™) and the squamous cell carcinoma cell lines 2106T and 2427T (in-house derived cell lines, [30]) were cultivated in DMEM/Ham’s F-12 (Thermo Fisher Scientific, Schwerte, Germany) with 10% fetal bovine serum (FBS, Thermo Fisher Scientific) and 1 × GlutaMAX™ (Thermo Fisher Scientific). For HCC827 (ATCC^®^ CRL-2868™, ADC), H2228 (ATCC^®^ CRL-5935™, ADC), and HCC15 (DSMZ ACC-496, SQCC), RPMI 1640 (Thermo Fisher Scientific) completed with 10% FBS was used. The complete growth medium for the SQCC cell line H1869 (ATCC^®^ CRL-5900™) was ACL-4 medium supplemented with 10% FBS. As this cell line in part remained in suspension and did not attach completely, the medium supernatant was included for all experimental approaches. For the ADC cell line A427, EMEM (Thermo Fisher Scientific) with 10% FBS was used. The patient-derived primary cell lines 4950T, 170162T, and 161735T were cultivated in serum-free DMEM/Ham’s F-12 with epithelial airway growth factors (Promocell, Heidelberg, Germany) and ROCK inhibitor (Rho-associated, coiled-coil containing protein kinase inhibitor; Stemcell Technologies, Cologne, Germany).

### 4.3. Transient Gene Knockdown by siRNA Transfection

Cells were seeded in 12-well plates at a density of 4–5 × 10^4^ cells/well and kept in the incubator at 37 °C and 5% CO_2_ overnight. For transfection, Lipofectamine™ RNAiMax (Thermo Fisher Scientific) was used according to the manufacturer’s protocol. The final siRNA concentration was 10 nM. Treated cells were incubated for 72 h at 37 °C and 5% CO_2_. The following siRNAs were used: targeting *FAM83A*: Hs_FAM83A_2 ccagaccgtcaagcacaacaa, Hs_FAM83A_3 ccggaacatcctctccaagtt, Hs_BJ-TSA-9_4 cagcgccactgtgtacttcca (Qiagen, Hilden, Germany); targeting *FAM83B*: SASI_Hs01_00046208 atagccttcccgtgatgag, SASI_Hs01_00046209 aactgaacaaccttgtttc, SASI_Hs01_00046210 attcgatgaggaagaatgc (Merck KGaA, Darmstadt, Germany).

For transfection approaches in 24-well plates, 2–2.5 × 10^5^ cells were seeded per well, and the protocol was adjusted according to the manufacturer’s instructions. As a control, AllStars Negative Control siRNA (Qiagen) was used. Efficiency was confirmed by quantitative Real-Time PCR, and suitable siRNAs were pooled.

### 4.4. Tyrosine Kinase Treatment

Cells were seeded in 24-well plates at densities of either 8 × 10^4^ cells/well or 2–3 × 10^4^ cells/well if they were transfected with siRNA prior to TKI treatment. The growth medium was replaced by serum-free medium for 16 h. Afterwards, the cells were treated with 10 µM, 1 µM, and 0.1 µM AZD9291 (ApexBio Technology, Houston, TX, USA) or erlotinib (Cell Signaling Technology, Frankfurt A.M., Germany) in growth medium for 24 h. As a control, growth medium with 0.1% DMSO was used. The experiments were repeated three times with three or four technical repeats each.

### 4.5. Total RNA Isolation and cDNA Synthesis

For RNA isolation from patient tumor tissue, a tumor content of ≥50% was the minimum prerequisite. A total of 10–15 tumor cryosections (10–15 µm) from each patient were sliced, and the first as well as the last section of a series were stained with H&E. A lung pathologist determined the proportion of viable tumor cells, stromal cells, healthy lung cells, and necrotic areas. Total RNA was isolated from patient tissue using an AllPrep DNA/RNA/miRNA Universal kit (Qiagen) according to the manufacturer’s instructions. An RNeasy Mini kit (Qiagen) was used to isolate RNA from the cell lines. Afterwards, the quality of total RNA was assessed by utilizing an Agilent 2100 Bioanalyser and an Agilent RNA 6000 Nano kit (Agilent Technologies, Boeblingen, Germany). With the Transcriptor First Strand cDNA Synthesis kit (Roche, Basel, Switzerland), total RNA was transcribed to complementary DNA and used for quantitative polymerase chain reaction (qPCR). A complete description of the procedure is provided elsewhere [31].

### 4.6. Quantitative Real-Time Polymerase Chain Reaction (qPCR)

Volumes of 5 µL cDNA (corresponding to 5 ng of isolated total RNA) were utilized for qPCR with the LightCycler480^®^ (Roche) in 384-well plates according to the Minimum Information for Publication of qPCR Experiments (MIQE) guidelines [32]. Universal ProbeLibrary (UPL) assay (Roche) was used as the amplification and detection system. Gene-specific primers (TIB Molbiol, Berlin, Germany) were combined with the primaQuant 2 × qPCR Probe-MasterMix (Steinbrenner Laborsysteme, Wiesenbach, Germany). Threshold cycle (C_t_) values were evaluated with the LightCycler480^®^ software release 1.5 and the 2nd derivative maximum method (Roche). For the comparison of gene expression in tumor and non-malignant samples, the relative expression of the genes (normalized to the housekeepers) was calculated (ΔC_t_ values). The same procedure was performed for the comparison in EGFR-wild-type and -mutant samples. The following primers and UPLs were used for the detection of *FAM83A* and *B, EGFR*, and the two housekeeper genes *ESD* and *RPS18*: FAM83A forward (UPL #12, 5′-CAGATCTCTGACAGTCACCTCAAG-3′), FAM83A reverse (UPL #12, 5′-CTGCCTGACTTGGCACAGTA-3′), FAM83B forward (UPL #20, 5′-GCATTCGGTGTCTTCATTAGC-3′), FAM83B reverse (UPL #20, 5′-TCTTGTCTTGTCTACCAAAAAGGTT-3′), EGFRv1 forward (UPL #50, 5′-ACACAGAATCTATACCCACCAGAGT-3′), EGFRv1 reverse (UPL #50, 5′-ATCAACTCCCAAACGGTCAC-3′), ESD forward (UPL #50, 5′-TCAGTCTGCTTCAGAACATGG-3′), ESD reverse (UPL #50, 5′-CCTTTAATATTGCAGCCACGA-3′), RPS18 forward (UPL #46, 5′-CTTCCACAGGAGGCCTACAC-3′), RPS18 reverse (UPL #46, 5′-CGCAAAATATGCTGGAACTTT-3′). The complete procedure is described elsewhere [31].

### 4.7. Proliferation Assay and Quantification of Cells by Lactate Dehydrogenase (LDH) Activity Measurement

Cells were quantified by measuring the LDH activity with a cytotoxicity detection kit according to the manufacturer’s instructions (Roche Diagnostics GmbH, Mannheim, Germany). Briefly, the cells that were previously treated with siRNA and/or tyrosine kinase inhibitors were lysed and centrifuged for 3 min at 13,000 rpm at room temperature. The supernatant was then further processed. The assay was repeated in three independent biological approaches using three or four technical repeats each.

### 4.8. Soft Agar Assay

One characteristic of tumor cells is their ability of anchorage independent growth. The soft agar assay was used to estimate this ability, e.g., after siRNA knockdown of a target gene (protocol adopted from Liu, F. [33]).

The cells, previously transfected in a 12-well plate, were harvested and diluted to a final concentration of 1 × 10^4^ cells/ml in complete culture medium. Then, 3 mL of the cell suspension was mixed with low-melt agarose (Bio&Sell GmbH, Feucht/Nuernberg, Germany) to a final agarose concentration of 0.36% and plated onto a bottom layer of 0.75% agarose in complete culture medium in 60 mm culture dishes. The plates were incubated at 37 °C and 5% CO_2_ for two to three weeks. The colonies were stained with 0.04% (*w/v*) crystal violet in 2% (*v/v*) ethanol in Dulbecco’s Phosphate-Buffered Saline (DPBS, Thermo Fisher Scientific) for 30 min to 1 h at room temperature. Colonies with diameters of at least 100 µm were then counted in 10 randomly selected squares which were engraved into the bottom of the dishes (Appendix A). The experiment was performed three times with two technical replicates per treatment.

### 4.9. Migration Assay

The in vitro tumor cell migration assay using ThinCerts™ (Greiner Bio-One GmbH, Frickenhausen, Germany) was adopted from Schneider, M. [34]. Cells were cultivated in 12-well plates and treated with the respective siRNAs. Thereupon, the growth medium was replaced by serum-free medium for 16 h at 37 °C and 5% CO_2_. The cells were then treated with 10 µg/mL mitomycin C (Merck KGaA) in 1 mL serum-free medium for 2 h in the incubator to inhibit further cell proliferation. Afterwards, 5 × 10^4^ cells in 300 µL serum-free medium were transferred onto the ThinCert™ which was placed into a 12-well plate containing 500 µL of full growth medium. After 24 h at 37 °C in a CO_2_ incubator, the cells that had migrated through the ThinCert™ were detached and quantified by LDH activity measurement. The experiment was performed three times with four technical repeats each to ensure reproducibility and reliable results.

### 4.10. Statistical Analyses

qPCR data were statistically analyzed using REMARK criteria [35] with SPSS 24.0 for Windows. The primary endpoint of the study was overall survival. The overall survival was calculated from the date of surgery until the last date of contact or death. Univariate analysis of survival data was performed according to Kaplan and Meier and using the Cox proportional hazards model. The cut-off between high and low expression was identified by CutOff Finder version 2.1 (Translational Tumor Research Team, Institute of Pathology, Charité—Universitätsmedizin Berlin; Appendix A). The log-rank test was used to test the significance of the differences between the groups. A *p*-value of less than 0.05 was considered significant. Multivariate survival analysis was performed using the Cox proportional hazards model. The non-parametric Mann–Whitney *U* test was used to investigate significant differences between the patient groups. For cell-based assays, unpaired t-test was performed. The Spearman ranked correlation coefficient test was performed for correlation analyses. Visualization of the qPCR data was made by GraphPad Prism 5.

## 5. Conclusions

With the data from qPCR, we could show that *FAM83A* and *FAM83B* are potential new diagnostic and prognostic biomarkers for distinct NSCLC subtypes. Further, their role seems to be highly regulated by different factors, including histology, mutational status, and EGFR signaling. Because of their high overexpression in tumor compared to non-neoplastic tissue, they might be interesting new targets in NSCLC personalized therapies and enhance the specificity and efficiency of treatments against cancer cells.

## Figures and Tables

**Figure 1 cancers-11-00652-f001:**
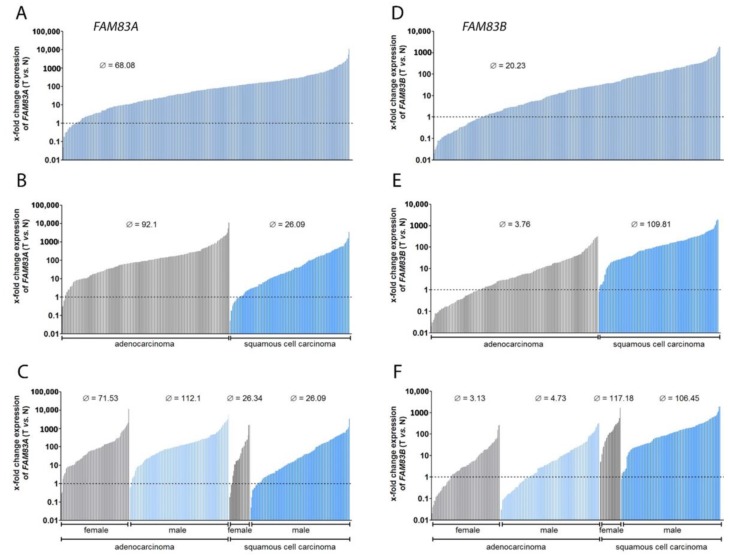
The FAMily with sequence similarity 83 (*FAM83*) genes *FAM83A* and *FAM83B* are highly overexpressed in non-small-cell lung cancer (NSCLC). The waterfall plots depict the x-fold change expression of *FAM83A* and *FAM83B* in tumor vs. normal tissue. (**A**) *FAM83A* expression in all patient samples, (**B**) separated by histology, (**C**) and furthermore by gender; (**D**–**F**) corresponding plots for *FAM83B*. Equal expression of the genes in tumor and adjacent non-neoplastic tissue was set to 1 (dotted line); Ø = median overexpression.

**Figure 2 cancers-11-00652-f002:**
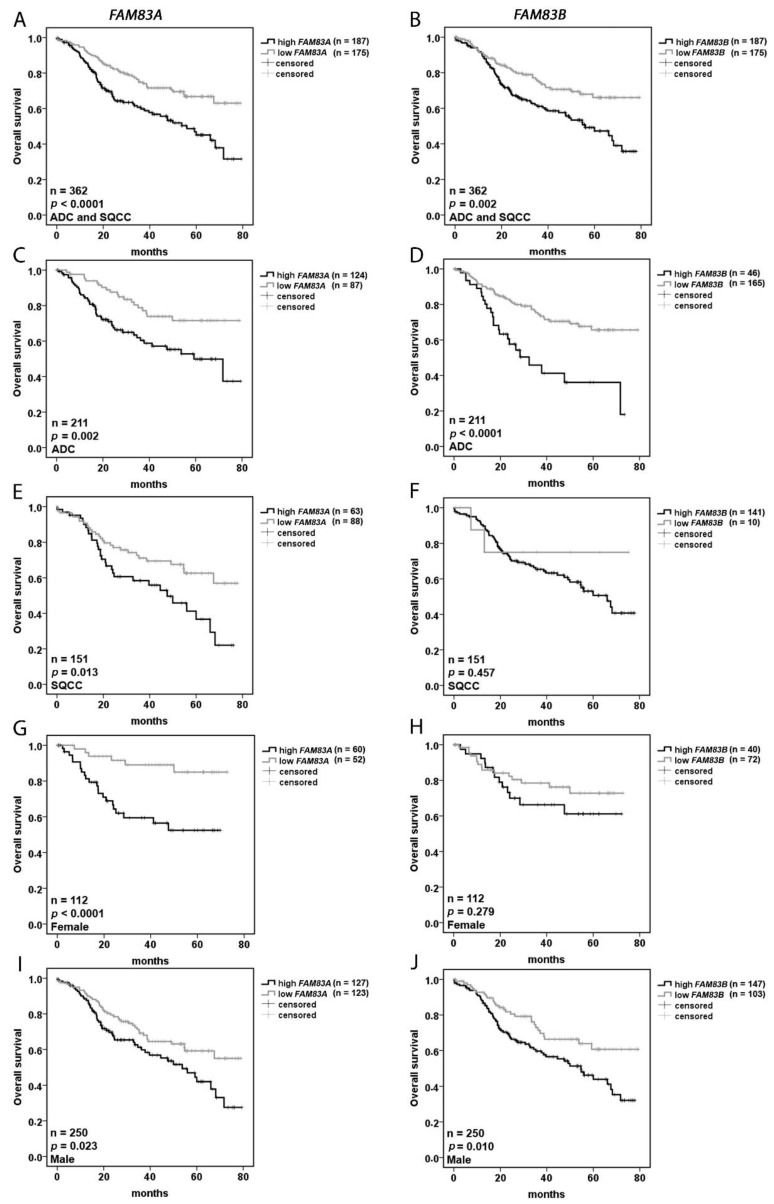
*FAM83A* and *FAM83B* expression has a prognostic value in NSCLC patients. Effect of (**A**) *FAM83A* and (**B**) *FAM83B* tumor expression on the overall survival of all patients, (**C**,**D**) of patients with ADC and (**E**,**F**) SQCC, and (**G**,**H**) of female and (**I**,**J**) male patients.; *p* < 0.05 was considered significant.

**Figure 3 cancers-11-00652-f003:**
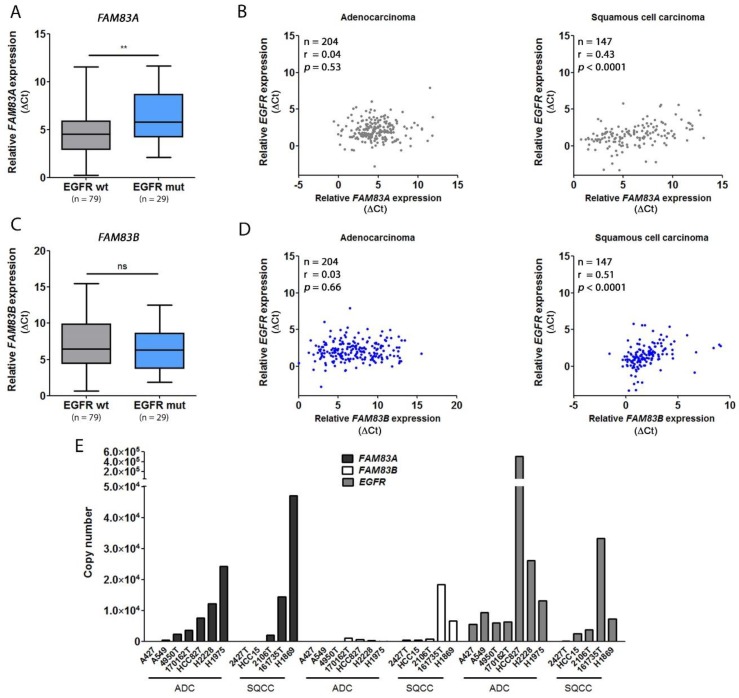
Relation between *FAM83A* and *B* expression and EGFR status and expression. (**A**) Relative expression of *FAM83A* in patients with wild-type EGFR and mutational variants normalized to the housekeeper genes *ESD* and *RPS18*. As relative expression is depicted (∆C_t_), lower values correspond to a higher expression. (**B**) Correlation plots between *FAM83A* and *EGFR* expression in the tumor tissue. (**C**,**D**) represent the respective results for *FAM83B*. (**E**) Absolute copy number of *FAM83A*, *FAM83B*, and *EGFR* in ADC and SQCC cell lines. The correlations were estimated using the Spearman’s rank correlation coefficient. For statistical analysis, Mann–Whitney *U* test was performed (** *p* < 0.005, ns = not significant).

**Figure 4 cancers-11-00652-f004:**
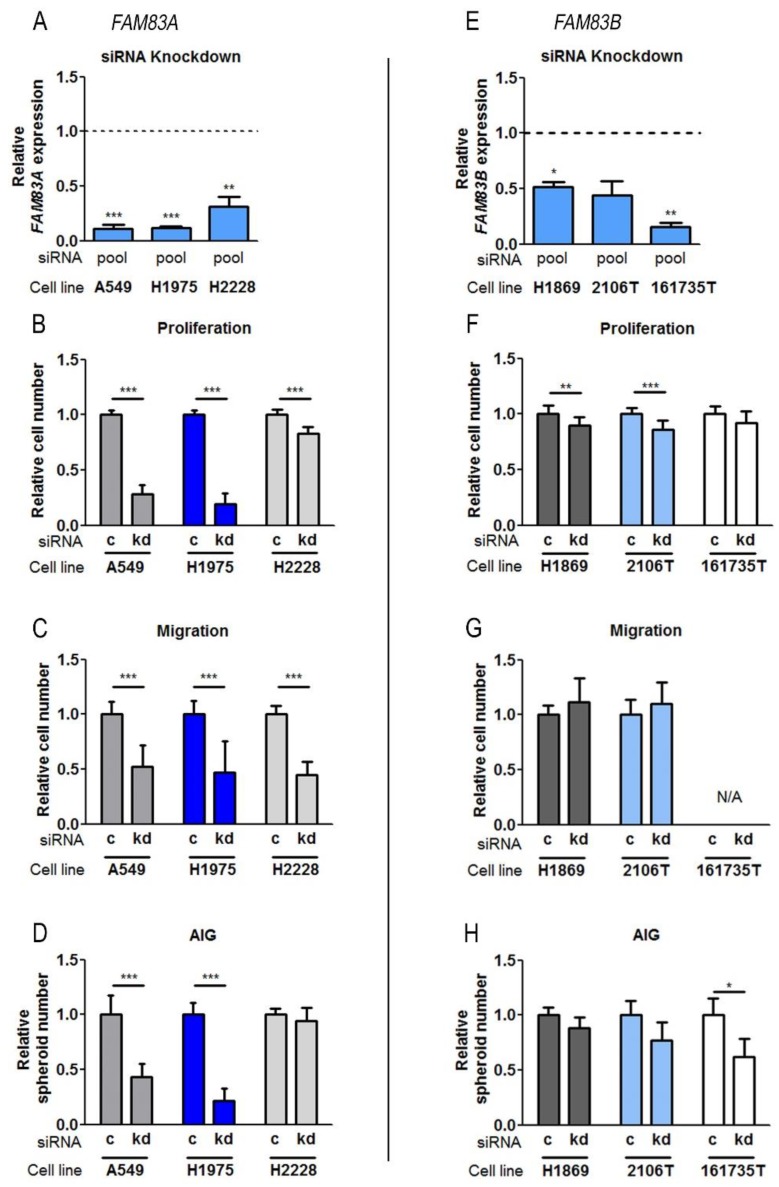
Investigation of the role of *FAM83A* and *B* in NSCLC cell lines. (**A**) *FAM83A* depletion efficiency. (**B**–**D**) Effect of decreased gene expression in ADC cell lines on (**B**) proliferation, (**C**) migration, and (**D**) ability of anchorage-independent growth (AIG). Relative cell or spheroid number of cells after *FAM83A* knockdown (kd) compared to control (c). (**E**–**H**) Corresponding results for *FAM83B*. At least three independent biological replicates were used. Significance was determined using the unpaired *t*-test (*** *p* < 0.0001, ** *p* < 0.005, * *p* < 0.05).

**Figure 5 cancers-11-00652-f005:**
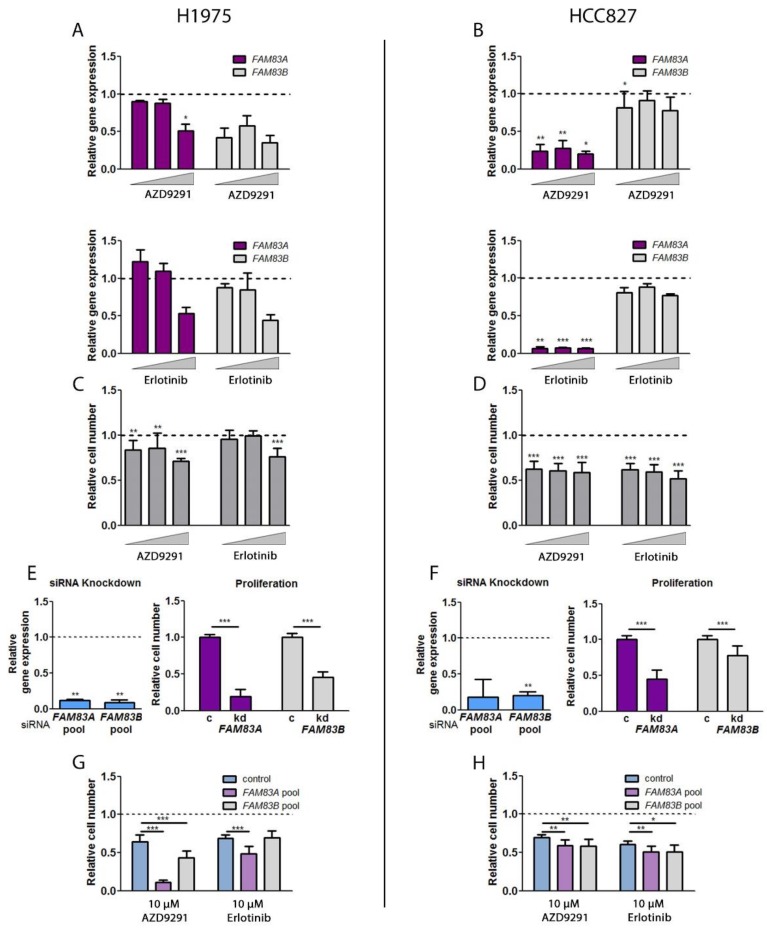
Relation between *FAM83A* and *B* and the EGFR pathway in ADC cell lines. (**A**) Gene expression of *FAM83A* and *B* after treatment with the tyrosine kinase inhibitors (TKIs) AZD9291 and erlotinib in H1975 and (**B**) in HCC827. The TKIs were applied at concentrations of 0.1 µM, 1 µM, and 10 µM for 24 h. (**C**) Effect of TKI treatment on cell viability in H1975 and (**D**) HCC827 cells. (**E**) siRNA-mediated depletion of *FAM83A* and *B* and its impact on cell proliferation in H1975 and (**F**) HCC827 (Relative cell number after *FAM83* knockdown (kd) compared to control (c)). (**G**) Influence of *FAM83A* and *B* depletion on TKI treatment in H1975 and (**H**) HCC827 cells. The dotted line at 1 represents either the expression (**A**,**B**,**E**,**F**,) or the cell number (**C**,**D**,**G**,**H**) in control-treated cells. At least three independent biological replicates were used. Significance of expression and cell viability variation were estimated using unpaired *t*-test (*** *p* < 0.0001; ** *p* < 0.005; * *p* < 0.05).

**Table 1 cancers-11-00652-t001:** Patient characteristics of the cohort investigated in the project.

Gene expression Analyses					
Parameter	*n*	(%)	Parameter	*n*	(%)
Median Age	65 (38–88)		P Stage		
Gender	362		IA	37	10
Male	250	69	IB	90	25
Female	112	31	IIA	70	19
Histology			IIB	51	14
Adeno	211	58	IIIA	105	29
Squamous	151	42	IIIB	9	2
Therapy			ECOG		
OP	212	59	0	320	88
OP/RT	13	4	1	32	9
OP/CT	100	28	2	4	1
OP/RT/CT	37	10	n.d.	8	2

OP: surgery, CT: chemotherapy, RT: radiotherapy, ECOG: Eastern Cooperative Oncology Group, n.d.: no data.

**Table 2 cancers-11-00652-t002:** Spearman correlation between *FAM83A* and *B* expression in the two main NSCLC histological types ADC and SQCC.

*FAM*	*FAM83B*	Histology	*p*-Value
*FAM83A*	*r* = 0.21	ADC	0.0025
*FAM83A*	*r* = 0.35	SQCC	<0.0001

Expression values from the tumor tissue were used for the analysis. A value of *r* = 1 depicts complete correlation.

**Table 3 cancers-11-00652-t003:** Univariate and multivariate analysis of *FAM83A* and *B*.

**Univariate Analysis**		
Variable (high vs. low)	Significance	Hazard Ratio (95% CI)
*FAM83A*	<0.0001	1.965 (1.366–2.817)
*FAM83B*	0.002	1.808 (1.252–2.612)
**Multivariate Analysis**		
Variable	Significance	Hazard Ratio (95% CI)
*FAM83A* (high vs. low)	<0.0001	1.980 (1.370–2.857)
Stage	<0.0001	1.076 (1.050–1.103)
Age	0.018	1.024 (1.004–1.045)
Sex (female vs. male)	0.062	0.664 (0.431–1.022)
Histology (ADC vs. SQCC)	0.391	0.853 (0.593–1.227)
*FAM83B* (high vs. low)	0.001	2.283 (1.393–3.745)
Stage	<0.0001	1.073 (1.047–1.100)
Age	0.018	1.024 (1.004–1.044)
Sex (female vs. male)	0.084	0.682 (0.442–1.052)
Histology (ADC vs. SQCC)	0.036	1.663 (1.034–2.672)

The expression of a gene is estimated significant with *p* < 0.05. The effect of overall survival was analyzed, including the variables *FAM83* expression (high vs. low), tumor stage, age, sex (female vs. male), and histology (adenocarcinoma, ADC, vs. squamous cell carcinoma, SQCC).

**Table 4 cancers-11-00652-t004:** Characteristics of the patients in the epidermal growth factor receptor (EGFR) cohort.

Gene Expression Analyses
Parameter	*n*	(%)	Parameter	*n*	(%)
Median Age	62 (40–81)		P Stage		
Gender	108		IA	12	11
Male	57	53	IB	34	31
Female	51	47	IIA	4	4
Histology			IIB	14	13
Adeno	108	100	IIIA	35	32
Therapy			IIIB	4	4
OP	52	48	IV	5	5
OP/RT	8	7	ECOG		
OP/CT	30	28	0	88	81
OP/RT/CT	18	17	1	17	16
EGFR Driver Mutation	29	27	2	2	2
Exon 19	14	13	n.d.	1	1
Exon 20	1	1			
Exon 21	14	13

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
