# Peer review of "FAM83A and FAM83B as Prognostic Biomarkers and Potential New Therapeutic Targets in NSCLC"

_cancers, 2019, doi:10.3390/cancers11050652_

Round 1

Reviewer 1 Report

It has been reported previously that FAM83A is overexpressed in lung adenocarcinoma and regulates MAPK Signaling in human cancer (PMID: 24736947) as well as that FAM83B is a biomarker for diagnosis and prognosis lung squamous cell carcinoma (n=113 patients) (PMID: 25586059). Previous reports also have demonstrated that FAM83A and FAM83B are oncogenic and confer resistance to EGFR-TKIs in breast cancer (doi:10.1172/JCI60517). In this study, Richtmann and co-authors have shown that FAM83A and FAM83B are overexpressed in non-small-cell lung cancer at RNA levels by qRT-PCR. They demonstrated that FAM83A and FAM83B are poor prognostic biomarkers in NSCLC. Further functional assay using pooled siRNAs against FAM83A or FAM83B has demonstrated that blocking FAM83A and FAM83B inhibited tumor cell proliferation. In addition, knocking down FAM83A or FAM83B enhanced cytotoxicity of NSCLC cell lines induced by EGFR tyrosine kinase inhibitors, erlotinib and AZD9291. However, there is no innovation in this study regarding to roles of FAM83A and FAM83B in NSCLC compared with previous published literatures. This study was lack of validation of both FAM83A and FAM83B at protein levels. In addition, approaches of testing functions of both FAM83A and FAM83B in mutant EGFR signaling pathway are not convincing. Methods using pooled siRNAs are unusual. My main concerns are as below.

1.       Many labels of Figure numbers in main text are confusing.

2.       In lines 134 - 137, the description contradicted with data shown in Figure 3A. The Figure 3A showed that expression levels of FAM83A in EGFR mutant lung adenocarcinoma patients were higher than those in EGFR wild type patients. 

3.       In lines 138 - 139, it does not make sense to perform correlation analysis between (total) EGFR expression at RNA levels and expression levels of FAM83A or FAM83B. The total EGFR expression levels do not reflect status of EGFR activation. To evaluate activated EGFR signaling pathways, protein levels of phosphorylated proteins, such as EGFR, ERK, and AKT should be evaluated.

4.       In Table 4, how to define “EGFR cohort”? What are the genotypes of EGFR wild type (79 patients among 211 adenocarcinoma patients)? Which genotypes were excluded in EGFR wild type except for mutant EGFR?

5.       In Line 176, at least two separated siRNAs rather than pooled siRNA should be used to test function of FAM83A and FAM83B.

6.       In lines 177 - 178, the authors mentioned that “It was not possible to confirm the knockdown by western blot due to a lack of specific commercial antibodies.”. It does not make sense. Sigma as well as other commercial companies provides specific antibodies to FAM83A and FAM83B.

7.       In lines 180 - 181, “depletion led to a highly significant decrease of all investigated properties.”. FAM83A expression level in A549 cells are very low compared to that in H1975 cells. Why knockdown of FAM83A could also decrease the proliferation, migration and other properties in FAM83A-low-expressing A549 cells?

8.       On page 11, in Figure D, it is not clear what the dot line represents and how long the treatments last. If the cells were treated for three days, 10 uM AZD9291 (very high concentration) could kill most of the H1975 cells. Thus, the cell viability of H1975 cells treated with AZD9291 (specially targeting T790M) shown in the first column in the Figure does not make sense.

9.       In lines 433 – 434, Methods – 4.9 Migration assay, does mitomycin C treatment affect expressions levels of FAM83A and/or FAM83B?

Author Response

Response to Reviewer 1 Comments

Point 1: Many labels of Figure numbers in main text are confusing.

Response 1: We agree and reorganized the labelling of figures 2 and 5 to simplify the structure of the experiments and the references in the text.

Point 2: In lines 134 - 137, the description contradicted with data shown in Figure 3A. The Figure 3A showed that expression levels of FAM83A in EGFR mutant lung adenocarcinoma patients were higher than those in EGFR wild type patients.

Response 2: In general, for qPCR analyses the gene expression (Ct value) is normalized to one or more housekeeper genes. This normalization results in a relative gene expression value (∆Ct). Genes which are highly expressed show a shorter distance to the housekeeper gene expression compared to genes with a lower expression. Consequently, lower values indicate a higher gene expression. The explanation can be found in the corresponding image caption: “As relative expression is depicted (∆Ct), lower values correspond to a higher expression.”

Point 3:   In lines 138 - 139, it does not make sense to perform correlation analysis between (total) EGFR expression at RNA levels and expression levels of FAM83A or FAM83B. The total EGFR expression levels do not reflect status of EGFR activation. To evaluate activated EGFR signaling pathways, protein levels of phosphorylated proteins, such as EGFR, ERK, and AKT should be evaluated.

Response 3: The aim of the correlation analyses was to gain a general insight into a possible relation of the different expression levels rather than the status of EGFR activation. This point was investigated by analysing the expression of FAM83A and B in EGFR wildtype and mutant ADC, therefore only the data of patients with driver mutations within the EGFR gene were used. However, your proposal of evaluating the EGFR signalling pathway might represent an important approach for future experiments. We already focused on this comment in our discussion but tried to point this out more in detail.

Point 4:   In Table 4, how to define “EGFR cohort”? What are the genotypes of EGFR wild type (79 patients among 211 adenocarcinoma patients)? Which genotypes were excluded in EGFR wild type except for mutant EGFR?

Response 4: This point was also addressed by another reviewer. Therefore, we added more information about this cohort in the Materials and Methods section 4.1, as well as in the main text (discussion section). Indeed, this cohort is an independent one and only partly included in the main cohort (Table 1).

Point 5:   In Line 176, at least two separated siRNAs rather than pooled siRNA should be used to test function of FAM83A and FAM83B.

Response 5: Prior to the knockdown experiments using the pooled siRNAs, single siRNAs were tested and analysed via qPCR (see supplementary material). siRNAs were pooled to reduce off-target effects (Hannus et al., 2014). Furthermore, the Cancers Instructions for Authors request to use at least two gene-specific siRNAs for gene silencing experiments (https://www.mdpi.com/journal/cancers/instructions accessed on 23.04.2019). Therefore, we believe that pooling of the siRNAs leads to a more specific effect.

Hannus, M. et al. (2014) ‘SiPools: Highly complex but accurately defined siRNA pools eliminate off-target effects’, Nucleic Acids Research, 42(12), pp. 8049–8061. doi: 10.1093/nar/gku480.

Point 6:   In lines 177 - 178, the authors mentioned that “It was not possible to confirm the knockdown by western blot due to a lack of specific commercial antibodies.” It does not make sense. Sigma as well as other commercial companies provides specific antibodies to FAM83A and FAM83B.

Response 6: We tested several antibodies against FAM83A and B which are commercially available. For example, we applied antibodies against FAM83A from Santa Cruz (Art. No. sc-86979) and Sigma-Aldrich (Art. No. SAB1103068) and against FAM83B from GeneTex (Art. No. GTX107222) in immunoblot analyses. Here, neither knockdown experiments nor approaches with overexpressed FAM83A or B in plasmid vectors could validate any specific signal for the antibodies. Furthermore, anti-FAM83B antibodies from Atlas Antibodies and Sigma-Aldrich were shown to be uncertain for western blot analyses, as well (https://www.proteinatlas.org/ENSG00000168143-FAM83B/antibody accessed on 24.04.2019; Antibodies HPA031464 and HPA031465). This problem has also been shown by  other groups that investigated FAM83A and B (Bufton et al., 2018). Additionally, we tried to produce anti-FAM83A and anti-FAM83B antibodies (Squarix Biotechnology). They did not show specific signals, either. As we could not validate any of the tested antibodies and due to inconsistent and insecure results, we decided not to perform or present any relevant experiments based on the application of antibodies.

Bufton, J. C. et al. (2018) ‘The DUF1669 domain of FAM83 family proteins anchor casein kinase 1 isoforms’, Science Signaling, 11(531), p. eaao2341. doi: 10.1126/scisignal.aao2341.

Point 7:   In lines 180 - 181, “depletion led to a highly significant decrease of all investigated properties.” FAM83A expression level in A549 cells are very low compared to that in H1975 cells. Why knockdown of FAM83A could also decrease the proliferation, migration and other properties in FAM83A-low-expressing A549 cells?

Response 7: This is indeed a very interesting point that first surprised us, as well. In the beginning we chose A549 for the experiments because the expression of FAM83A was shown to be relatively low compared to other cell lines. However, depletion of the gene had a significant impact on cellular functions. Despite the lower FAM83A expression, an absolute FAM83A copy number of over 500 copies in 5 ng cDNA was still measured for A549 (Figure 3 E). This value represents sufficient and measureable amounts of RNA. However, the results indicate that FAM83A is highly regulated in the cells and that downregulation of the gene leads to severe effects even in cells with lower expression values. We tried to clarify your comment in our discussion.

Point 8:   On page 11, in Figure D, it is not clear what the dot line represents and how long the treatments last. If the cells were treated for three days, 10 uM AZD9291 (very high concentration) could kill most of the H1975 cells. Thus, the cell viability of H1975 cells treated with AZD9291 (specially targeting T790M) shown in the first column in the Figure does not make sense.

Response 8: We apologize for the missing information. The explanation of the dotted line was added to the image caption (see manuscript Figure 5). The information about the duration of TKI treatment is mentioned in the Materials and Methods section 4.4 (Tyrosine kinase treatment): “Afterwards, cells were treated with 10 µM, 1 µM and 0.1 µM AZD9291 (ApexBio Technology, Houston, TX, USA) or erlotinib (Cell Signaling Technology, Frankfurt a.M., Germany) in growth medium for 24 h.” With regard to your comment, we also added this information in the figure legend.

Point 9:   In lines 433 – 434, Methods – 4.9 Migration assay, does mitomycin C treatment affect expressions levels of FAM83A and/or FAM83B?

Response 9: As mitomycin C acts as a cytostatic, it probably influences numerous genes which are involved in cell proliferation and cell growth. In our experimental setup, mitomycin C was applied on FAM83A and B depleted cells as well as negative control-treated cells. Thus, possible bias can be excluded. The investigation of any direct effect of the cytostatic on FAM83A and/or B expression was not the focus of our current study.

Reviewer 2 Report

It is an interesting retrospective analysis looking at new prognostic biomarkers  such as the oncogenes FAM83 A and B in Non Small Cell Lung Cancer There are correlations between histology  and prognostic  value of the expression of the two oncogenes which needs to be elucidate in a prospective study and the authors have to comment thi s point. It has been reported a number of observations  in several cell lines and some of the reports are in conflict with the reported absence of interactions  with mutant Egfr status. In lung cancer , monoclonal antibodies have very limited activity against Egfr TKIs and thi should be reported.  The possibility that FAM83 A and B can be involved in primary resistanc to TKIs shuld be addressed. With these comments  I consider the manuscript well written and original and reccomend it's publication.

Author Response

Response to Reviewer 2 Comments

Point 1: It is an interesting retrospective analysis looking at new prognostic biomarkers  such as the oncogenes FAM83A and B in Non-Small Cell Lung Cancer There are correlations between histology  and prognostic value of the expression of the two oncogenes which needs to be elucidate in a prospective study and the authors have to comment this point.

Response 1: Since our current study focused on surgical patients with stages I-III, a prospective study could investigate the role of the genes especially in stage IV patients. We agree to your comment and explained the relevance and prognostic value of FAM83A and B in late stages more in detail in our discussion.

Point 2: It has been reported a number of observations in several cell lines and some of the reports are in conflict with the reported absence of interactions with mutant EGFR status.

Response 2: Patient-derived results might differ from findings in cell culture based experiments. For example, other mutations in the tissue might influence FAM83A and B expression. Due to the tumor heterogeneity it cannot be prevented that signals from other cells (e.g. immune cells, fibroblasts) might affect gene expression values. As described in the materials and methods section, we only used tumor tissue with a tumor content > 50 % to reduce these interfering effects. Nevertheless, we cannot exclude any variety in FAM83A and B gene expression. Here, the cell lines are used as models to identify possible relations of FAM83A and B and EGFR and their biological function in NSCLC.

As another point and regarding patients with a mutant EGFR status, patients who undergo surgery are not tested for molecular alterations when the tumor is completely resected. Only surgical patients developing metastasis at a later time or patients with recurrence of tumor disease are retrospectively analyzed for EGFR mutations. Consequently, sufficient tissue amounts of patients with EGFR mutations are rarely available and our numbers for statistical analysis need to be increased in future studies. We addressed this point in the discussion section.

Point 3: In lung cancer, monoclonal antibodies have very limited activity against EGFR TKIs and this should be reported.

Response 3: We are sorry, but it is not clear for us which point the reviewer would like to address with this comment. Maybe, the reviewer wants to say that monoclonal antibodies are not as effective as specific TKIs. We checked the related literature but could not find any review demonstrating a clear benefit of one or the other therapy. This depends on different factors, like the kind of mutation, the generation of the TKI, the smoking status but also pharmacokinetic variability. Since we are not sure what was meant with the comment, we could not consider it.

Point 4: The possibility that FAM83A and B can be involved in primary resistance to TKIs should be addressed.

Response 4: We already addressed this point in the introduction, the results and the discussion. Therefore, we believe that this function of FAM83A and B is already clearly described in our manuscript.

Reviewer 3 Report

The manuscript entitled “FAM83A and FAM83B as prognostic biomarkers and potential new therapeutic targets in NSCLC” by Richtmann et al focused on FAM83A and FAM83B as novel diagnostic prognostic and potential predictive biomarkers in NSCLC patients. The manuscript was well written and suitable for publication after revisions:

Major revisions:

- Although the Authors correctly stated in the Introduction section that NSCLC is typically diagnosed in advanced stages, the percentage of patients with IIIB or IV stages was very low (9/362, 2%, n = 9 IIIB and n = 0 IV in the analyzed population Table 1; and 9/108, 9%, n = 4 IIIB and n = 5 IV in the EGFR cohort Table 4). These data could affect the results of the study. Could the Authors discuss this issue?   

- The Authors analyzed n = 362 patients as shown in Table 1. Then n = 108 ADC patients were analyzed to evaluate the relation between expression level of FAM83A and B and EGFR in vivo. Are these two different population? Could the Authors better define the study population in material and methods section?

- The Authors treated H1975 cell line (L858R/T790M+) with both erlotinib and osimertinib. The presence of EGFR exon 20 p.T790M gives resistance to first generation TKIs (such as erlotinib). Why the Authors adopted these association? Could the Authors better discuss this point?

- In the Material and Method section page 13 lines 331-332: why the Authors did not use the 2015 WHO classification? Could the Author better explain this point?

Minor revisions:

- In the Introduction section page 2 line 45-47 (Non-small cell lung cancer accounts for 85 % of all lung cancers and is further classified into the main histological groups squamous cell carcinoma (SQCC) and adenocarcinoma (ADC)): could the Author mention also large cell neuroendocrine carcinoma (LCNEC)?

- In the Introduction section page 2 line 64: please use the acronym “tyrosin kinase inhibitor”.

In the Results section page 10 lines 204-206 (Erlotinib is a reversible small molecule ATP analogue which has been found to be most effective in advanced NSCLC patients with a tumor containing exon 19 deletions or a L858R mutation in exon 21 of the gene encoding the EGFR kinase domain.): could the Authors provide a reference for this statement?

- In the Discussion section page 11 line 246: please use only the acronym FAM83.

- In the Discussion section page 12 line 251: please use the acronym for “adenocarcinoma”.

- In the Discussion section page 12 line 295: please use the acronym for “anchorage-independent growth”.

- The Authors should re-write the genes in italics.

Author Response

Response to Reviewer 3 Comments

Point 1: Although the Authors correctly stated in the Introduction section that NSCLC is typically diagnosed in advanced stages, the percentage of patients with IIIB or IV stages was very low (9/362, 2%, n = 9 IIIB and n = 0 IV in the analyzed population Table 1; and 9/108, 9%, n = 4 IIIB and n = 5 IV in the EGFR cohort Table 4). These data could affect the results of the study. Could the Authors discuss this issue?

Response 1: The patient cohort depicted in Table 1 differs from the EGFR cohort (see response 2). In general, patients with stage IV NSCLC do not undergo surgery. Exceptions would be patients who are initially staged as stage III, but are corrected to stage IV due to findings during surgical intervention. However, these cases are rare and were therefore not included in the presented patient cohort (n = 362) for statistical reasons.

When the tumor is completely resected, patients are normally not tested for EGFR mutations. This is only the case for surgical patients developing metastasis or for patients with tumor recurrence, who are then retrospectively analyzed. This leads to the circumstance that sufficient tissue amounts of patients with EGFR mutations are hardly available. Therefore, regarding the EGFR cohort we included surgical stage IV patients, as well, to increase the patient number. To clarify this point, we added more information to our discussion section.   

Point 2: The Authors analyzed n = 362 patients as shown in Table 1. Then n = 108 ADC patients were analyzed to evaluate the relation between expression level of FAM83A and B and EGFR in vivo. Are these two different population? Could the Authors better define the study population in material and methods section?

Response 2: This point was also addressed by another reviewer. We therefore added more information in the Materials and Methods section 4.1.

Point 3: The Authors treated H1975 cell line (L858R/T790M+) with both erlotinib and osimertinib. The presence of EGFR exon 20 p.T790M gives resistance to first generation TKIs (such as erlotinib). Why the Authors adopted these association? Could the Authors better discuss this point?

Response 3: In breast cancer cells FAM83A and B are related to resistance mechanisms, inter alia, regarding TKIs. Therefore, in the above mentioned experiment we focused not only on EGFR inhibition but also on the general influence of the different TKIs on FAM83A and B expression. To clarify this point, we provided additional information in the respective section 2.5.

Point 4: In the Material and Method section page 13 lines 331-332: why the Authors did not use the 2015 WHO classification? Could the Author better explain this point?

Response 4: All tissue samples included in the presented study were collected between 2003 and 2015 and therefore diagnosed after the 2004 WHO classification. We added this information in the Materials and Methods section 4.1.

Minor revisions: The Authors should re-write the genes in italics.

We controlled the manuscript and checked that all genes are written in italics. Regarding FAM83A and B some statements are related to the proteins, not the genes (e.g. discussion section, line 305 “sufficient amounts of FAM83B”; discussion section, lines 323-324 “the involvement of FAM83A and B in EGFR signalling”).

Besides, all other comments were corrected in the manuscript.

Round 2

Reviewer 1 Report

In Figures 3A and 3C, to make it more understandable, the y axis tile may use either “ΔCt” or “fold change”. I am satisfied with the rest parts of the revised manuscript.

Author Response

According to your comment, we adapted the title of the y-axes in the noted figure. The new labelling of the axes hopefully clarified the intention of these graphs. Additionally, we added further information to the figure legend.

We would like to thank you for your constructive comments during the review process.

Reviewer 3 Report

I have not further comments.

Author Response

The reviewer had no further comments. We thank you for your constructive revision of our manuscript.